# Microstructure Reconstruction and Multiphysics Dynamic Distribution Simulation of the Catalyst Layer in PEMFC

**DOI:** 10.3390/membranes12101001

**Published:** 2022-10-14

**Authors:** Zhigang Zhan, Hao Song, Xiaoxiang Yang, Panxing Jiang, Rui Chen, Hesam Bazargan Harandi, Heng Zhang, Mu Pan

**Affiliations:** 1State Key Laboratory of Advanced Technology for Materials Synthesis and Processing, Wuhan University of Technology, Wuhan 430070, China; 2Foshan Xianhu Laboratory of the Advanced Energy Science and Technology Guangdong Laboratory, Xianhu Hydrogen Valley, Foshan 528200, China; 3School of Automotive Engineering, Wuhan University of Technology, Wuhan 430070, China; 4State Key Laboratory of Geomechanics and Geotechnical Engineering, Institute of Rock and Soil Mechanics, Chinese Academy of Sciences, Wuhan 430071, China

**Keywords:** proton exchange membrane fuel cell, catalyst layer microstructure, nano-ct, reconstruction, heat and mass transport, electrochemical reaction, proton conduction, dynamic processes

## Abstract

Due to the complexity of both material composition and the structure of the catalyst layer (CL) used in the proton-exchange membrane fuel cell (PEMFC), conjugated heat and mass transfer as well as electrochemical processes simultaneously occur through the CL. In this study, a microstructure model of CL was first reconstructed using images acquired by Nano-computed tomography (Nano-CT) of a real sample of CL. Then, the multiphysics dynamic distribution (MPDD) simulation, which is inherently a multiscale approach made of a combination of pore-scale and homogeneous models, was conducted on the reconstructed microstructure model to compute the corresponded heat and mass transport, electrochemical reactions, and water phase-change processes. Considering a computational domain with the size of 4 um and cube shape, this model consisting of mass and heat transport as well as electrochemical reactions reached a stable solution within 3 s as the convergence time. In the presence of sufficient oxygen, proton conduction was identified as the dominant factor determining the strength of the electrochemical reaction. Additionally, it was concluded that current density, temperature, and the distribution of water all exhibit similar distribution trends, which decrease from the interface between CL and the proton-exchange membrane to the interface between CL and the gas-diffusion layer. The present study not only provides an in-depth understanding of the mass and heat transport and electrochemical reaction in the CL microstructure, but it also guides the optimal design and fabrication of CL components and structures, such as improving the local structure to reduce the number of dead pores and large agglomerates, etc.

## 1. Introduction

Renewable energy sources such as solar energy and wind energy are unstable and intermittent during generation, and thus these valuable electric energies are difficult to apply continuously and stably. The employment of PEMFC combined with hydrogen production from water electrolysis may greatly improve the utilization rate and stability of renewable energy [1]. The PEMFC consists of a bipolar plate (BPP), proton-exchange membrane (PEM), catalyst layer (CL), microporous layer (MPL), and gas-diffusion layer (GDL). The role of PEM is to conduct protons from the anode to the cathode, and the most commonly used PEM is Nafion series membranes which are perfluorinated sulfonic acid membranes containing both hydrophobic PTFE backbone and hydrophilic sulfonic acid group [2,3]. Due to its important functions and complicated structure, CL has already been one of the most important components in PEMFC [4]. However, it is difficult to characterize the heat and mass transport as well as electrochemical reaction processes in the CL via visualization and experimental in-situ methods. Therefore, establishing numerical microstructure models of CL is of great significance to understand and capture the complex processes in it.

Nano-Computed Tomography (Nano-CT) technique can provide lossless observation images of three-dimensional microstructures. Shi et al. [5] used Nano-CT to characterize the three-dimensional micro-crack structure of low-volatile coal and simulated the permeability characteristics of single-phase water flow in the micro-crack network using COMSOL. Zhao et al. [6] studied the pore structure characteristics of coal via Nano-CT, discussed the computational fluid dynamics simulation method based on the micro-nano pore structure, and then carried out the permeability simulation of the reconstructed microstructure. The results showed that the permeability of coals is low and anisotropic in all three different directions. Cetinbas et al. [7] reconstructed the PEMFC cathode catalyst layer (CCL) containing high surface-area porous carbon (HSC) using Nano-CT and simulated the effects of HSC morphology, agglomerate structure and relative humidity on the local oxygen transport resistance, concluding that smaller size or lower aspect ratio agglomerates facilitate better O_2_ transfer to the catalyst surface at high relative humidity. Braaten et al. [8] studied the degradation process of the PEMFC CCL by Nano-CT characterization and discovered that the combination of HSC support and PtCo catalyst was able to limit the loss of ECSA and Pt band formation in the CL better than the combination of Vulcan and pure Pt. Cui et al. [9] used Nano-CT to reconstruct the PEMFC CCL to study how various dimensions carbon materials affect the electrochemical performance of catalyst coated membrane (CCM). The outcomes showed that increasing the electrochemical performance of CCM by using carbon nanotubes in the CCL with low Pt loading is a successful strategy.

In addition to the experimental characterization of Nano-CT for microscopic reconstruction of CLs, numerical simulations are effective methods to study the structure and composition on the performance of PEMFCs. The conventional macroscopic homogeneous model can study the effect of flow channel structures and humidification conditions on cell performance. However, the relationship between Pt loading, Nafion content and cell performance is not considered in the macroscopic model [10,11,12]. The agglomerate model establishes the relationship between the content of CL and the cell performance. Sun et al. [13] built a two-dimensional CL polymer electrolyte coated catalyst particle agglomerate model which was used for fuel cell cathode half-cell simulations. The results showed that the catalyst utilization in the cathode agglomerates was extremely low at high current densities due to the limited catalysts. By modeling CL agglomerate and one-dimensional PEMFC model, Sohn et al. [14] carried out numerical simulations of CCL and concluded that cell with smaller pore in CCL exhibits more concentration loss at high current densities. Hosseini et al. [15] performed simulations of two-dimensional open-anode PEMFC utilizing an agglomerate model. According to the simulation results, a rise in relative humidity promotes water’s diffusion from the cathode to the anode. Cosse et al. [16] proposed and validated a transient PEMFC agglomerate model for predicting the peak current of double-layer capacitor, and the simulation results showed that the gas supply stoichiometry has little effect on the absolute peak of the short-circuit current. Yang et al. [17] investigated the effect of CL on the PEMFC cold start using an agglomerate model and concluded that a larger I/C decreases CL porosity and agglomerate pore size, which significantly reduces the critical ice fraction of cold start failure.

Although the agglomerate model considers the relationship between CL component content and cell performance, the effects of CL pore structure and solid phase structure on cell performance and lifetime have not been deeply studied. Therefore, it is significant to establish a mesoscopic scale model based on experimental characterization data to study the distribution of Multiphysics and enhance the performance of PEMFCs. In recent decades, the study of mesoscopic scale reconstruction [18,19] and simulation [20,21,22,23,24,25,26] in PEMFCs mainly focuses on the GDL. This is because there are nanoscale pore structure and complex electrochemical reaction processes in CL, the resolution of CL reconstruction is required to be higher, and the simulation process is more complicated. Babu et al. [27] used Nano-CT to reconstruct Pt-free catalysts and simulated the CL proton conduction under different Nafion content. Compared with the experimental results, it was shown that higher Nafion content could easily result in CL flooding and then affect proton conduction, which ultimately leads to the decline of fuel cell performance under high current density. Cetinbas et al. [28] used Nano-CT to reconstruct the agglomerate structure with secondary pores in CL, meanwhile, the microstructure with primary pores, Carbon and Pt particle distribution was numerically reconstructed based on Transmission Electron Microscopy (TEM) and Scanning electron microscope (SEM) characterization images. The two reconstructed structures were combined to form a computational domain, and the electron, proton and oxygen transport properties were investigated by using STAR CCM+. Satjaritanun et al. [29] investigated the transport properties in different scale components of the PEMFC through a hybrid model. Firstly, the microstructures of the gas-diffusion media were obtained by Nano-CT, then the transport phenomena in the membrane electrode assembly (MEA) was studied via Lattice Boltzmann Agglomeration Method. Meanwhile, the data calculated by Lattice Boltzmann Agglomeration was exchanged with the flow channel, and conventional CFD method was used to simulate the transport phenomena in flow channel. Finally, the factors of affecting the cell performance were studied.

In this study, the three-dimensional microstructure of the CL was reconstructed by Nano-CT experimental data, and the pore structural characteristics and gas-diffusion media tortuosity were analyzed. The Knudsen effect was considered to correct the transport properties, and a pore-scale model for heat and mass transport as well as electrochemical processes was built. For current research status, i.e., that Nano-CT cannot easily distinguish the Pt, carbon particles, and Nafion, the solid phase region was treated with a homogeneous method, and the multiphysics dynamic distribution characteristics of the CL were studied. The comprehensive simulation methods and results presented in this study not only demonstrate the importance of transport properties in CL, but also help in advancing the in-depth understanding of electrochemical reaction processes in PEMFC.

## 2. Microstructure Reconstruction and Physical Properties Correction of CL

### 2.1. Sample Preparation

In this paper, the MEA, consisted of a Nafion 211 membrane and 60% mass fraction Pt/C industrial catalyst, which was produced by Wuhan University of Technology Hydrogen Power Technology Co. Ltd. The Pt loading and active area of the MEA were 0.4 mg/cm^2^ and 25 cm^2^. The treatments of MEA were as follows: firstly, the GDL was peeled off from both sides of the MEA and then the CCM was cut into a 2 mm × 2 mm square. Secondly, the treated CCM sample was immersed in saturated cesium sulfate solution (CsSO_4_) with deionized water for 78 h to exchange H^+^ with Cs^+^. Finally, to avoid sulfate deposition, the samples were shocked with deionized water for 5 min and dried to remove excess water.

### 2.2. Nano-CT Imaging

Nano-CT imaging was performed at the 4W1A Topography Imaging Experiment Station of Beijing Synchrotron Radiation Facility. The sample was analyzed through Nano-CT with a resolution of 50 nm in the Zernike phase contrast mode of 8 keV and the exposure time was 10 s. A series of images were obtained when the sample was rotated at 0.3° between −70° and +70°, then the 2D images were corrected and 3D reconstruction was processed by Xradia XMReconstructor software. Finally, a computational domain of 4 μm × 4 μm × 3 μm was selected from the reconstructed microstructure, and histogram equalization and normalization were performed on it to enhance the contrast. The volume was binarized into solid and pore spaces using artificial thresholding, as shown in Figure 1; there were some isolated pore structures called dead pores.

### 2.3. Structural Characterizations of Reconstructed Model

The porosity of the reconstructed CL shown in Figure 1 was 32.4%, as obtained by Avizo software. It is consistent with the results obtained by the mercury intrusion method (MIP) experimental method, which indicates the feasibility and validity of the reconstruction method. Figure 2a,b show the pore size distribution and tortuosity distribution of the CL, respectively. The average pore size and average tortuosity of the reconstructed CL were 136 nm and 2.1, which is similar to ref. [28].

The three parameters of transport properties can be used to correct the CL physical properties in the subsequent section.

### 2.4. Correction of Transport Parameters in CL

The relative gas-diffusion coefficient $D_{i}{}^{R}$ in the CL is the ratio of effective diffusion coefficient $D_{i}{}^{eff}$ to the molecular diffusion coefficient $D_{i}{}^{bulk}$ as listed in Table 1. The Knudsen effect is neglected, it can be expressed in terms of porosity ε and tortuosity τ as follows [30]:(1)$$D_{i}{}^{R}=\frac{D_{i}{}^{eff}}{D_{i}{}^{bulk}}=\frac{\varepsilon }{\tau }$$
where the subscript i can represent the gas species of air in the cathode: oxygen, nitrogen, and water vapor at the cathode.

The gas-diffusion coefficient $D_{i}$ in the CL is related to the Knudsen diffusion coefficient $D_{i}^{k}$ and the molecular diffusion coefficient $D_{i}^{bulk}$, it can be solved by the Bosanquet equation [31]:(2)$$\frac{1}{D_{i}}=\frac{1}{D_{i}^{k}}+\frac{1}{D_{i}^{bulk}}$$

When the Knudsen effect is considered, the relative diffusion coefficient Equation (1) is modified as follows:(3)$$D_{i}{}^{R}=\frac{D_{i}{}^{eff}}{D_{i}{}^{bulk}}=\frac{\varepsilon }{\tau }\frac{1}{\left(1+\frac{D_{i}{}^{bulk}}{D_{i}{}^{k}}\right)}$$
(4)$$D_{i}{}^{k}=\frac{2}{3}{\left(\frac{8RT}{\pi M_{i}}\right)}^{1/2}r$$
where R is the universal gas constant, T is the temperature, r is the average pore radius, $M_{i}$ is the relative molecular mass.

The effective electrical conductivity $\kappa_{ele}^{eff}$ and effective proton conductivity $\kappa_{ion}^{eff}$ of the reconstructed CL are solved as follows:(5)$$\kappa_{ele}^{eff}=\sigma_{ele}V^{ele}/\tau^{ele}$$
(6)$$\kappa_{ion}^{eff}=\sigma_{ion}V^{ion}/\tau^{ion}$$
where $\sigma_{ele}$ is the electron conductivity, $\sigma_{ion}$ is the proton conductivity, $V^{ele}$ and $V^{ion}$ are the carbon volume fraction and ionomer volume fraction of the CL, respectively. $\tau^{ele}$ and $\tau^{ion}$ are the tortuosity of carbon and ionomer, respectively.

Based on the above equations, the gas diffusivity and conductivity in the reconstructed CL were corrected. Figure 3a shows the relationship between relative diffusion coefficient and porosity in the CL with and without Knudsen diffusion. The relative diffusion coefficient in the CL corrected by porosity and tortuosity is smaller than that corrected by Bruggmen. Additionally, the relative diffusion coefficient of the CL considering Knudsen diffusion was the smallest; Knudsen diffusion had a great effect on the relative diffusion coefficient. The above results of the relative diffusion coefficient correction are similar to ref. [35].

Figure 3b–d show the relationship between the effective diffusion coefficients and porosity in the CL for oxygen, water vapor, and nitrogen with and without Knudsen diffusion, respectively. The gas’s effective diffusion coefficient in the CL increased with the porosity, which is due to gas transport resistance decreasing with the porosity. Additionally, the gas’s effective diffusion coefficient in the CL when Knudsen diffusion is considered was smaller than those without Knudsen diffusion. In addition, the effective diffusion coefficient of the gas species obtained from the Bruggmen equation was relatively high. Figure 3e,f show the relationship between effective electron/proton conductivity and carbon/ionomer volume fraction, respectively. Both the effective electron and proton conductivity in the CL increased with the carbon/ionomer volume fraction. This is because more carbon particles and ionomer provide more pathways for electrons and protons to transport, respectively.

## 3. Mathematical Analysis of Reconstructed CL

### 3.1. Geometric Models

The above three-dimensional reconstructed microstructure of CL was imported into the Multiphysics simulation software COMSOL as shown in Figure 4. The pore phase and solid phase of the CL are represented in red and blue, respectively.

In this study, Nafion, C and Pt were not distinguished in the solid phase; they were divided according to the proportion obtained by the MEA experimental characterization.

### 3.2. Mathematical Model

There are complicated processes such as heat and mass transport, phase change, and electrochemical reaction in the CL of PEMFC. To simplify the calculation, the following assumptions were made in this study [36]:

(1) All the gas species are ideal gases.

(2) The water generated by the electrochemical reaction in the CL is ionomer water, and then diffuses toward the area of low concentration. The ionomer water can convert to water vapor when it reaches the pore/solid phase interface.

(3) The liquid water at the boundary can be completely removed in time.

(4) The heat, mass transport and electrochemical reactions are calculated using a homogenous model due to the difficulty in distinguishing various components of the solid phase and the amount of calculation in this study.

Based on the above assumptions, the mathematical model in this study includes the following governing equations [31,37]:

The mass conservation equation:(7)$$\frac{\partial }{\partial t}((1-s_{lq})\rho_{g})+\nabla \cdot (\rho_{g}\vec{u})=S_{mass}$$
where $s_{lq}$, $\rho_{g}$, $\vec{u}$ and $S_{mass}$ represent the liquid water saturation, mixed gas density, mixed gas velocity vector and mass source term, respectively.

Inertial and viscous forces were ignored because they have little effect on performance in the model; the momentum conservation equation was simplified to:(8)$$\;(1-s_{lq})\vec{u}=-\frac{K_{g}}{\mu_{g}}\nabla p_{g}$$
where $K_{g}$, $\mu_{g}$, and $p_{g}$ represent mixed gas permeability, viscosity, and gas pressure, respectively

The energy conservation equation is:(9)$$\frac{\partial (c_{p}\rho^{eff}T)}{\partial t}+\nabla \cdot (\vec{u}c_{p}\rho^{eff}T)=\nabla \cdot (K^{eff}\nabla T)+S_{Q}$$
where $c_{p}$, $K^{eff}$, $\rho^{eff}$, and $S_{Q}$ represent the constant pressure heat capacity, temperature, effective thermal conductivity, effective density, and energy source terms, respectively.

The component conservation equation is:(10)$$\frac{\partial }{\partial t}((1-s_{lq})\rho_{g}Y_{i})+\nabla \cdot (\rho_{g}{\vec{u}}_{g}Y_{i})=\nabla \cdot (\rho_{g}D_{i}^{eff}\nabla Y_{i})+S_{i}$$
where Yi, $D_{i}^{e}{}^{ff}$, and $S_{i}$ represent the component concentration, the effective diffusion coefficient of the species *i*, and the component source term, respectively. The values of $D_{i}^{eff}$ are corrected after considering the Knudsen effect according to the method in the previous section.

The liquid water conservation equation is:(11)$$\frac{\partial (s_{lq}\rho_{lq})}{\partial t}+\nabla \cdot (\frac{K_{lq}\mu_{g}}{K_{g}\mu_{lq}}\rho_{lq}{\vec{u}}_{g})=\nabla \cdot (\rho_{lq}D_{lq}\nabla s_{lq})+S_{lq}$$
where $\rho_{lq}$ and $\mu_{lq}$ represent the density and viscosity of liquid water, $K_{lq}$ represents the liquid phase permeability, and $K_{g}$ represents the gas phase permeability. The liquid water diffusion coefficient $D_{lq}$ after considering the effect of capillary pressure difference can be expressed as:(12)$$D_{lq}=-\frac{K_{lq}}{\mu_{lq}}\frac{dp_{c}}{ds_{lq}}$$
(13)$$p_{ca}=\sigma \cos\theta {\left(\frac{\varepsilon }{K_{0}}\right)}^{0.5}\times [1.42(1-s_{lq})-2.12{\left(1-s_{lq}\right)}^{2}+1.26{\left(1-s_{lq}\right)}^{3}]\;\theta <90^{\circ }$$
(14)$$p_{ca}=\sigma \cos\theta {\left(\frac{\varepsilon }{K_{0}}\right)}^{0.5}\times [1.42s_{lq}-2.12s_{lq}{}^{2}+1.26s_{lq}{}^{3}]\;\theta >90^{\circ }$$
where $p_{ca}$, θ, and σ are capillary pressure, contact angle, and surface tension, respectively.

Charge conservation equation:(15)$$\nabla \cdot (\kappa_{ele}^{eff}\nabla \phi_{ele})+S_{ele}=0$$
(16)$$\nabla \cdot (\kappa_{ion}^{eff}\nabla \phi_{ion})+S_{ion}=0$$
where $\phi_{ele}$ is the electrical potential, $\phi_{ion}$ is the proton potential, $\kappa_{ele}^{eff}$ is the effective electrical conductivity, $\kappa_{ion}^{eff}$ is the effective proton conductivity, and $S_{ele}$ and $S_{ion}$ are the electron current source term and the proton current source term, respectively. Only the electrochemical reaction of cathode was considered in this study:(17)$$S_{ele}=J_{c}\cdot Ac$$
(18)$$S_{ion}=-J_{c}\cdot Ac$$
(19)$$J_{c}=j_{c,ref}^{v}{\left(\frac{C_{O_{2}}}{C_{O_{2},ref}}\right)}^{\gamma }\times (e^{-\frac{\alpha F}{RT}\eta_{c}})$$
where $J_{c}$ is the cathode electrochemical reaction rate, Ac is the active specific surface area, $j_{c,ref}^{v}$ is the cathode reference exchange current density, $C_{O_{2},ref}$ is the oxygen reference concentration, γ is the concentration index, $\eta_{c}$ is the cathode overpotential, α is the transport coefficient, and F is the Faraday constant. The subscript c indicates the cathode. The phase change equation can be referred to the literature [38].

### 3.3. Boundary Conditions and Initial Values

The model parameters and boundary conditions used in the model are listed in Table 2 and Table 3, respectively.

Since the computational domain of this study is part of the CL, the setting of the boundary conditions was determined by the PEMFC operating conditions and the location. In this study, the boundary conditions are given according to the literature [35] and were set as initial values as shown in Table 3.

### 3.4. Numerical Calculation Method

The number of grids in this model is 840,000, it has little effect on the calculation results when the grid number changes by 10%. The physical properties of the CL and the gas-diffusion coefficient were corrected after considering the tortuosity, porosity, and Knudsen effect. Finally, COMSOL Multiphysics 5.3a software was used to solve the governing equations of dynamic distribution of Multiphysics.

## 4. Results and Discussion

### 4.1. Oxygen Concentration Distributions

Figure 5a,b show the molar concentration distribution of oxygen in the pore phase at 1 s and 3 s, respectively. In the CL, diffusion is regarded as the only mode of oxygen transport. The oxygen concentration gradually decreased from the CL/GDL interface to the CL/PEM interface due to the consumption of electrochemical reaction. The difference of oxygen concentration between the two interfaces was 0.4 mol/m^3^ at 1 s and 0.6 mol/m^3^ at 3 s, respectively. In particularly, there are some dead pores in the computational domain where oxygen cannot enter as shown in Figure 5a,b. Therefore, the O_2_ concentration in dead pores is constant to zero, and no electrochemical reaction occurs in there. Figure 5c shows the average O_2_ molar concentration along the CL thickness direction at different moments. The difference between the maximum and minimum values of O_2_ molar concentration in the CL gradually became larger over time. The differences were 0.35 mol/m^3^, 0.4 mol/m^3^, 0.47 mol/m^3^, 0.6 mol/m^3^, and 0.61 mol/m^3^ at 0.5 s, 1 s, 2 s, 3 s, and 4 s, respectively. At 3 s and 4 s, the difference was almost unchanged, indicating that the electrochemical reaction in CL tended to be stable.

### 4.2. Current Density Distributions

Figure 6 is the contour of the current density distribution calculated by Equation (19) at six different moments. With the progress of the reaction, the electrochemical reaction gradually proceeded from the CL/PEM interface to the CL/GDL interface, and the current density always decreased from the CL/PEM interface to the CL/GDL interface. Because the oxygen concentration was 10.1 mol/m^3^ and water vapor concentration was 0 mol/m^3^ at the initial moment in this study, it is difficult for the ionomer in the CL to conduct protons; the oxygen reduction reaction starts at the CL/PEM interface where protons are sufficient. The water generated in CL gradually increased and diffused from the CL/PEM interface to the CL/GDL interface, which led to the increase of proton conductivity in the solid phase. However, there was a gradient of current density from the CL/PEM interface to the CL/GDL interface. This phenomenon indicates that the proton conduction is the dominant factor that affects the electrochemical reaction when the oxygen is sufficient, which are consistent with ref. [36]. The current density distribution in the calculation domain during 3–4 s did not change obviously, and it was in a dynamic equilibrium state.

In addition, the current density was larger in the solid phase region near the pore phase. This is because the oxygen diffusion coefficient in the CL pore phase is much larger than that in the solid phase, and there is sufficient oxygen in the solid phase region near the pore, which leads to the more intense electrochemical reaction and larger current density at this pore/solid interface. To sum up, the oxygen diffusion and the intensity of the electrochemical reaction are all restricted in the large solid phase due to the lack of sufficient contact area with the pores.

### 4.3. Temperature Distributions

Figure 7a shows the average temperature distribution of the CL/GDL interface, the CL/PEM interface, and the CL middle section (along the *Z*-axis direction) within 4 s. The temperature increased over time for all three different sections. Additionally, the temperature of the CL/PEM interface was the highest, followed by the CL middle section, and the lowest temperature was at the CL/GDL interface. After 3 s, the temperature of all the three interfaces did not increase, indicating that the electrochemical reaction became stable. Figure 7b shows the contour of temperature distribution on different sections along the CL thickness direction when the reaction was stable, which is similar to the current density distribution in Figure 6. Besides, the temperature was higher in the solid phase region near the pore. The overall temperature in the CL increased by 0.6 K from the beginning of the reaction to stability.

### 4.4. Liquid Water and Membrane Water Distribution

Figure 8a shows the profiles of the average liquid water saturation calculated by Equation (11) along the *Z*-axis at different moments. The liquid water saturation gradually decreased from the CL/PEM interface to the CL/GDL interface. This is because the electrochemical reaction firstly generated water at the CL/PEM interface and the water gradually diffused to the GDL/CL interface. The average liquid water saturation in the CL gradually increased with time, and the liquid water saturation profiles at 3 s and 4 s basically overlapped, indicating that the reaction was basically stable at 3 s. Additionally, the distribution of liquid water saturation is consistent with the distribution of current density. Figure 8b shows the distribution of the average liquid water saturation in the y = 0 section when the reaction was stable, which is consistent with the curve at 3 s in Figure 8a.

Figure 8c shows the profiles of the average ionomer water content in CL over time, assuming that the initial ionomer water content λ is 6. The water generated by electrochemical reaction diffused toward the region with a lower water concentration. Because the electrolyte in the CL absorbs water vapor and reaches saturation, the ionomer water did not increase any more after 3 s, and the stable average ionomer water content in the CL was around 17.5.

Figure 8d is the iso-surface of ionomer water along the *Z*-axis direction of CL when the reaction was stable, which is the same as the distribution of liquid water at the same time. It can be found that the distribution of ionomer water is hilly and undulating on any iso-surface due to the complex pore structure, the strength of electrochemical reactions, and the wettability of the ionomer inside the CL.

The distribution of oxygen, current density, temperature, and water in the computational domain tended to be stable after 3 s. Considering the dimensions of the computational domain, this transition time from the beginning of reaction to the steady state is basically consistent with the conclusion of ref. [40], which indicates the feasibility and validity of the study results.

## 5. Conclusions

In this study, a multiscale approach was developed to incorporate the effective transport properties computed from 3D reconstructed CL with a macroscopic model created in COMSOL Multiphysics 5.3a. In the first step, a three-dimensional microstructure geometry of CL was reconstructed using Nano-CT images of a commercial sample as well as its experimental data. In the following step, the distribution of pore sizes and tortuosity was analyzed. In the pore phase, the transport properties were first corrected by the Knudsen effect, and then the COMSOL Multiphysics 5.3a was used to simulate the dynamic processes of heat, mass transport, and electrochemical reaction in the CL. The results are listed as follows:

(1) It can be seen that the relative diffusion coefficient and effective diffusion coefficient of the gas in CL were smaller after considering Knudsen diffusion, indicating that Knudsen diffusion had a significant effect on the gas’s effective diffusion coefficient.

(2) Due to the difficulties in distinguishing multiple components of CL in the solid phase and the desire to reduce the computational cost, a Multiphysics simulation method was proposed in which the mesoscopic-scale structure was combined with the pore-scale model and the homogeneous model.

(3) It is evident that current density, temperature, and water distribution all exhibited similar distribution trends, decreasing from the CL/PEM interface to the CL/GDL interface.

(4) Under the introduced boundary and operating conditions, the mass transport, heat transport, and electrochemical reactions reached a stable solution after 3 s as the convergence time. Upon completion of this 3 s, oxygen concentration, current density, temperature, and water content no longer changed.

The simulation results help to understand the mass and heat transport and electrochemical reaction in the CL microstructure. It has guiding significance for the optimal design and fabrication of porous structures, such as improving the local structure to reduce the number of dead pores and large agglomerates, etc. Future research should focus on developing more precise reconstruction methods to realize the differentiation of the CL’s components, and then simulate and study the influence of components and structure on the electrochemical performance of the CL.

## Figures and Tables

**Figure 1 membranes-12-01001-f001:**
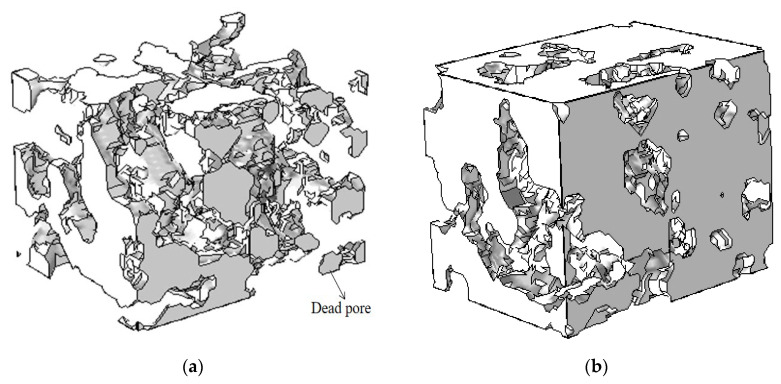
CL microstructure reconstruction. (**a**) Pore phase; (**b**) solid phase.

**Figure 2 membranes-12-01001-f002:**
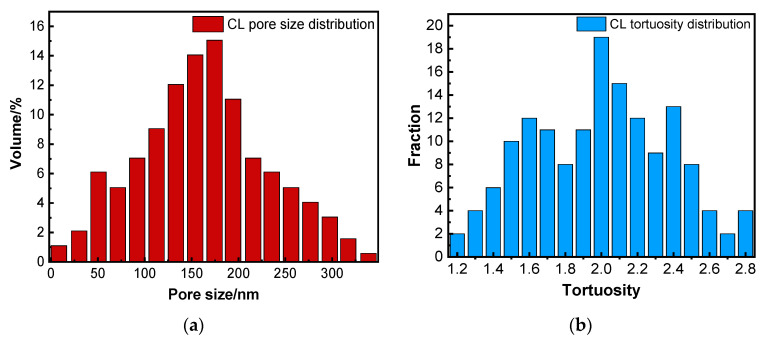
(**a**) Pore size distribution and (**b**) tortuosity distribution of CL.

**Figure 3 membranes-12-01001-f003:**
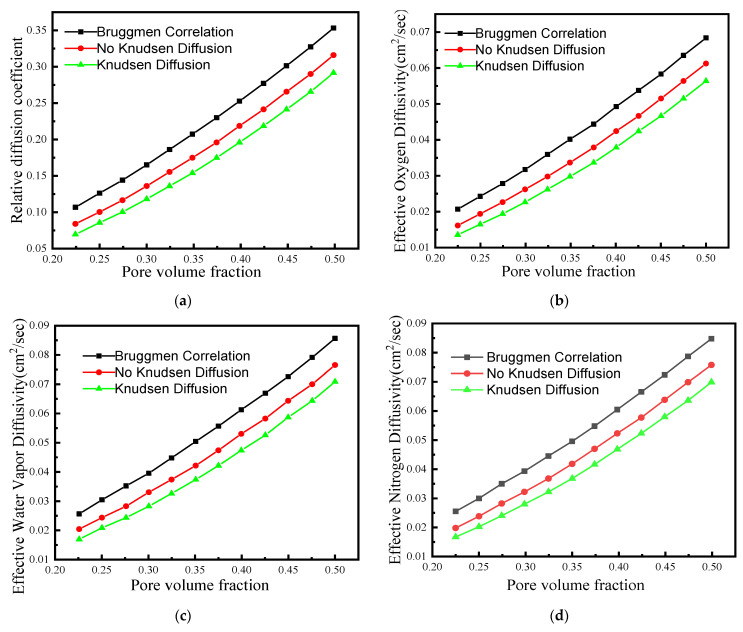
The relationship between various variables and porosity: (**a**) relative diffusion coefficient, (**b**) effective oxygen diffusivity, (**c**) effective water vapor diffusivity, (**d**) effective nitrogen diffusivity, (**e**) the relationship between the effective electron conductivity and carbon volume fraction, and (**f**) the relationship between the effective proton conductivity and ionomer volume fraction.

**Figure 4 membranes-12-01001-f004:**
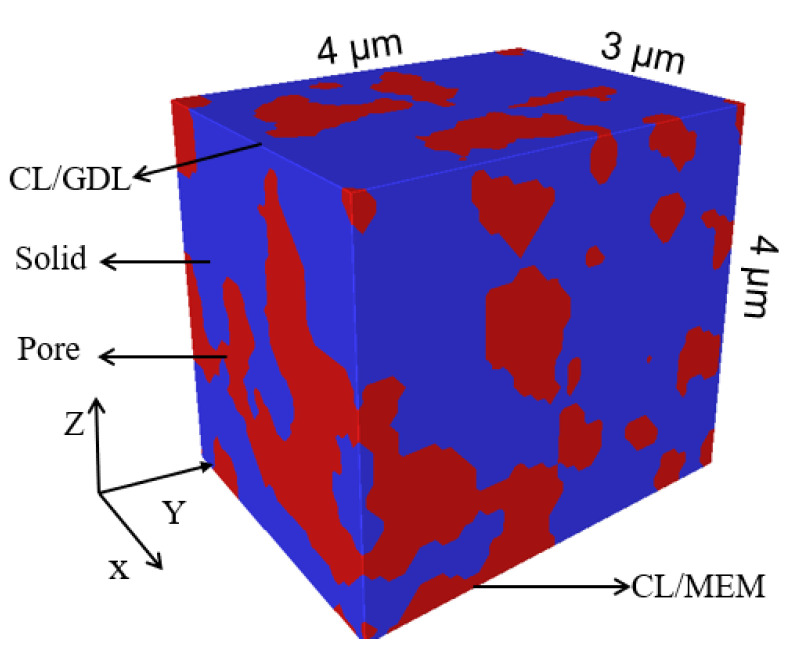
Geometric model of microstructure of CL.

**Figure 5 membranes-12-01001-f005:**
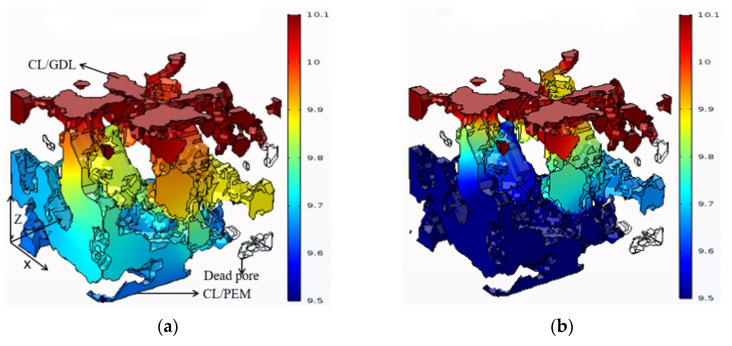
Distribution of oxygen molar concentration at different moments: (**a**) 1 s, (**b**) 3 s, and (**c**) oxygen molar concentration along the thickness of CL at different moments.

**Figure 6 membranes-12-01001-f006:**
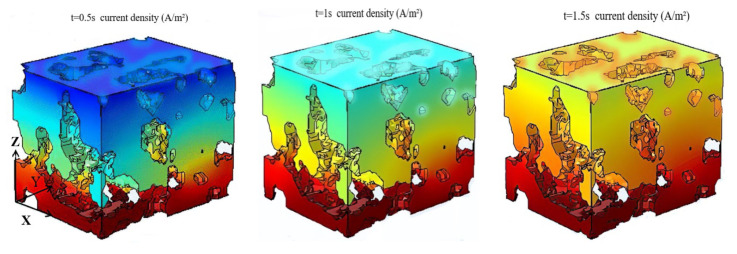
Distribution of current density in CL at different moments.

**Figure 7 membranes-12-01001-f007:**
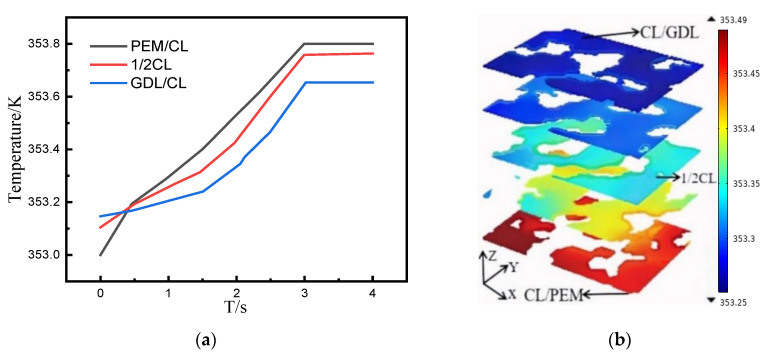
(**a**) Profiles of temperature at CL/GDL interface, CL/PEM interface, and CL middle section at different moments. (**b**) Temperature distribution of different sections along the CL thickness direction when the reaction is stable.

**Figure 8 membranes-12-01001-f008:**
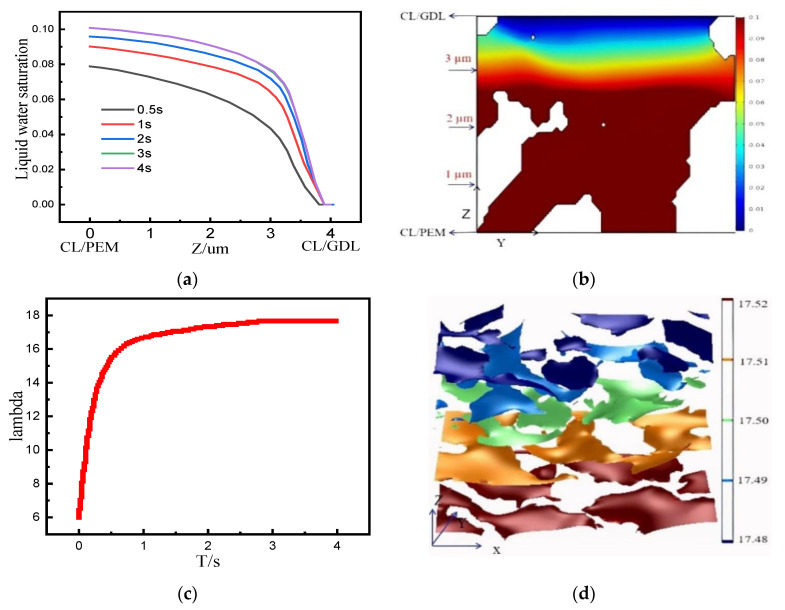
(**a**) The average liquid water saturation along the CL thickness direction at different moments. (**b**) Distribution of the average liquid water saturation on the y = 0 μm section when the reaction is stable. (**c**) Profiles of average ionomer water in the CL over time. (**d**) The iso-surface map of ionomer water along the thickness direction of the CL.

**Table 1 membranes-12-01001-t001:** Transport parameters used in the model [32,33,34].

Parameters	Values
$D_{O_{2}}^{bulk}$ (cm^2^/s)	0.129
$D_{H_{2}O}^{bulk}$ (cm^2^/s)	0.166
$D_{N_{2}}^{bulk}$ (cm^2^/s)	0.161
$\sigma_{ion}$ (S/cm)	0.02
$\sigma_{ele}$ (S/cm)	10.0

**Table 2 membranes-12-01001-t002:** Model parameters [35,39].

Parameters	Value
$j_{c,ref}^{v}$ (A/cm^2^)	1.659 × 10^−6^
α	0.61
$C_{O_{2},ref}$ (mol/cm^3^)	40.96 × 10^−6^
γ	1.0
F (C/mol)	96,487.0
R (J/(mol K))	8.314

**Table 3 membranes-12-01001-t003:** Boundary conditions and initial values.

Variables	PEM Side	GDL Side
Oxygen concentration (mol/cm^3^)	/	10.1 × 10^−6^
Liquid water saturation	/	0
Water vapor concentration (mol/cm^3^)	/	0
Proton potential (V)	0.908	0.9
Electron potential (V)	1.738	1.73
Temperature (K)	353.0	353.15

## Data Availability

Data is contained within the article.

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
