# Peer review of "Microstructure Reconstruction and Multiphysics Dynamic Distribution Simulation of the Catalyst Layer in PEMFC"

_membranes, 2022, doi:10.3390/membranes12101001_

Round 1
Reviewer 1 Report
The present study explored the material composition and structure of catalyst layer (CL) in the proton exchange membrane fuel cel. The heat, mass transport and electrochem- 15 ical reaction processes are as described as being complicated. In the study, the microstructure 16 of CL is reconstructed based on Nano-computed tomography (Nano-CT). Then, the Multiphysics dynamic distribution in the reconstructed CL is performed using a hybrid method of pore-scale model and homogenous model which considers mass and heat transport, electrochemical reaction and phase change processes. The results show that the mass, heat transport and electrochemical reaction process tend to be stable within 3s in the computational domain of 4um cube shape. The proton conduction is the dominant factor of the strength of the electrochemical reaction when oxy gen is sufficient. The investigation provides indepth understanding method in dynamic multiphysics distribution of CL. Below are some recommendations to improve the quality of the study.
1. In the abstract like 23 the wording should be 'trend'. Kindly change it.
2. The abstract should be re - written to highlight the exact contribution for the study in a form of a quantitative recommendation or qualitative.
3. In the introduction, some of the references are too old and goes as far back as 2011 and 2012. Authors are thus encouraged to rather use recent references at least in the last five years.
4. The work is novel and would really support the fuel cell research community but the authors should provide a vivid justification for how their developed model was validated.
5. The results though discussed should be compared with other published dated and the percentage differences should be clearly explained when submitting the revised manuscript.
6. How did the authors settle on the geometrical parameters for their model. Can the provide a reason for why these geometrical parameters were used in their study.
7. Due to the high volume of equations, authors must provide a nomenclature showing the Greek letters, subscript, superscript etc.
8. All equations in the manuscript should be properly referenced and cited in the main text.
9. All abbreviations and equations should be defined in the first instance.
10. The conclusion should equally be re - written to capture future studies in details.
Reviewer 2 Report
This manuscript explains the reconstruction and multiphysics dynamic distribution simulation of the catalyst layer in PEMFC. The manuscript clearly shows the mass, heat transport and electrochemical reaction process tend to be stable within 3s in the computational domain of 4um cube shape. The distribution tends of current density, temperature and water distribution are the same that decrease from the interface between CL and proton exchange membrane to the interface between CL and gas diffusion layer. The study provides a somewhat in-depth understanding method in dynamic Multiphysics distribution of CL and guides the optimal design and fabrication of CL components and structures. But there is some improvement in the English language and typos required. This manuscript can be accepted after minor improvement in grammatical typological errors.
Reviewer 3 Report
The manuscript reported the reconstructed microstructure of the catalyst layer (CT) based on Nano-CT. The Multiphysics dynamic distribution in the reconstructed CL is performed using a hybrid method of pore-scale model and homogenous model. It is found that proton conduction is the dominant factor in the strength of the electrochemical reaction when oxygen is sufficient. The distribution tends of current density, temperature, and water distribution to decrease from the interface between CL and proton exchange membrane to the interface between CL and gas diffusion layer. The study provides an in-depth understanding method in dynamic multiphysics distribution of CL and guides the optimal design and fabrication of CL components and structures.
I consider the content of this manuscript will meet the reading interests of the readers of the Membranes journal. However, there are certain English spelling and grammar issues, and also the discussion and explanation should be further improved. I suggest giving a minor revision and the authors need to clarify some issues or supply some more experimental data to enrich the content.
The detailed comments can be found in the PDF file.
